# Effect of vaccination on household transmission of SARS-CoV-2 Delta variant of concern

Frederik Plesner Lyngse [1,2,3 ✉], Kåre Mølbak [3,4], Matt Denwood [4], Lasse Engbo Christiansen [5], Camilla Holten Møller[3], Morten Rasmussen[3], Arieh Sierra Cohen[3], Marc Stegger [3], Jannik Fonager [3], Raphael Niklaus Sieber [3], Kirsten Maren Ellegaard[3], Claus Nielsen[3] & Carsten Thure Kirkeby[4]

Effective vaccines protect individuals by not only reducing the susceptibility to infection, but also reducing the infectiousness of breakthrough infections in vaccinated cases. To disentangle the vaccine effectiveness against susceptibility to infection ($VE_S$) and vaccine effectiveness against infectiousness ($VE_I$), we took advantage of Danish national data comprising 24,693 households with a primary case of SARS-CoV-2 infection (Delta Variant of Concern, 2021) including 53,584 household contacts. In this setting, we estimated $VE_S$ as 61% (95%-CI: 59-63), when the primary case was unvaccinated, and $VE_I$ as 31% (95%-CI: 26-36), when the household contact was unvaccinated. Furthermore, unvaccinated secondary cases with an infection exhibited a three-fold higher viral load compared to fully vaccinated secondary cases with a breakthrough infection. Our results demonstrate that vaccinations reduce susceptibility to infection as well as infectiousness, which should be considered by policy makers when seeking to understand the public health impact of vaccination against transmission of SARS-CoV-2.

---

[1] Department of Economics & Center for Economic Behaviour and Inequality, University of Copenhagen, Copenhagen, Denmark. [2] Danish Ministry of Health, Copenhagen, Denmark. [3] Statens Serum Institut, Copenhagen, Denmark. [4] Department of Veterinary and Animal Sciences, Faculty of Health and Medical Sciences, University of Copenhagen, Copenhagen, Denmark. [5] Department of Applied Mathematics and Computer Science, Dynamical Systems, Technical University of Denmark, Richard Petersens Plads, 324, DK-2800 Kgs. Lyngby, Denmark. ✉email: fpl@econ.ku.dk

The current COVID-19 pandemic caused by the SARS-CoV-2 virus is of major concern worldwide, and vaccination is a central part of the strategy to control the pandemic. Nevertheless, pandemic control is being challenged by the ability of SARS-CoV-2 to continuously evolve into new genomic variants of concern (VOC) with differing characteristics in terms of transmissibility and immune evasion. During 2021, the Delta VOC (SARS-CoV-2 Lineage B.1.617.2) became a major concern, due to the evidence for higher transmissibility compared to previous variants[1–5]. Furthermore, some studies suggest that the Delta VOC possess increased immune evasion properties relative to previous variants, causing a higher number of breakthrough infections in vaccinated individuals[6,7]. During the second half of 2021, the Delta VOC caused new surges in SARS-CoV-2 infections and hospitalizations in several countries, putting a pressure on health care systems despite broad vaccination roll-out. In response, several countries reintroduced restrictions and non-pharmaceutical interventions (NPIs) to sustain epidemic control. To understand the full potential of the Delta VOC, and choose the optimal mitigation strategy, it is essential to estimate the vaccine effectiveness for this variant.

There are several issues that make it difficult to estimate real-world vaccine effectiveness (VE). In principle, an infection with SARS-CoV-2 generally requires three things: (i) an infected case that is able to transmit the virus, (ii) a potential secondary case that is susceptible to infection, and (iii) a contact/transmission event between the infected case and the potential secondary case. Vaccinations can affect the susceptibility to infection of the exposed persons, but may also affect the infectiousness of the infected case with a breakthrough infection overcoming the effect of vaccination against infection. This makes it difficult to empirically separate the estimates of vaccine effectiveness against infection in exposed contacts ($VE_S$) from the estimates of vaccine effectiveness of infectiousness in primary cases ($VE_I$). A solution to obtain these estimates is to acquire secondary information on infected primary cases that allows them to be linked to their exposed contacts, and accounting for the vaccination status of both primary cases and exposed contacts in a statistical analysis.

In this study we used Danish nationwide administrative data to estimate vaccine effectiveness within households. We estimate two key parameters for the SARS-CoV-2 Delta VOC in Denmark in the period June–October 2021: Vaccine effectiveness against susceptibility to infection in household contacts ($VE_S$) and vaccine effectiveness against infectiousness in primary cases ($VE_I$).

## Results

Table 1 shows the summary statistics of the data included in this study. The study included 24,693 primary cases, of which 33% (8262) were fully vaccinated and 67% (16,431) were not vaccinated, and 53,584 household contacts, of which 49% (26,098) were fully vaccinated and 51% (27,486) were not vaccinated. The secondary attack rate (SAR) was generally lower among vaccinated contacts than among unvaccinated (Table 1). Comirnaty comprised 83% of the vaccinations in vaccinated individuals. No children below age 12 years were vaccinated during our study. Additional summary statistics are presented in Tables S1 and S2, and Fig. S6.

Table 2 shows the vaccine effectiveness (VE) estimates. The pooled vaccine effectiveness against susceptibility of infection in household contacts ($VE_S$)—unconditional on the vaccination status of the primary case—was 61% (95%-CI: 59–63). The $VE_S$ was 61% (95%-CI: 59–63) when the primary case was unvaccinated, compared to 46% (95%-CI: 40–52) when the primary case was fully vaccinated. The pooled vaccine effectiveness against infectiousness in primary cases ($VE_I$)—unconditional on the

vaccination status of the contact—was 42% (95%-CI: 39–45). The $VE_I$ was 31% (95%-CI: 26–36) when the contacts were unvaccinated, compared to 10% (95%-CI: 0–18) when the contacts were fully vaccinated. The total vaccine effectiveness ($VE_T$) was 66% (95%-CI: 63–68), i.e., when both the primary case and contact were fully vaccinated compared to both of them being unvaccinated. Note that the VE estimates across columns are not directly comparable as they are estimated on stratified samples.

Next, we estimated the pooled $VE_S$, pooled $VE_I$, and $VE_T$ for each combination of age group of the primary case and household contacts (Fig. 1). Generally, there was a positive VE across all age group combinations, implying that the SAR is reduced by vaccination of both the exposed and infected individuals. Also, there was generally a decreasing VE with age of both the primary case and contact.

Unvaccinated secondary cases have a significantly higher viral load (lower Ct value) compared to vaccinated secondary cases, independent of the day of testing after identification of the primary case (Fig. 2). Unvaccinated secondary cases have a lower Ct value of 1.6, which corresponds to a 3-fold higher viral load (this is calculated due to the doubling property of Ct measurements as $2^{1.6} = 3$) (Table S4).

Lastly, we performed a number of supplementary analyses to support our main results. We found evidence that age composition is important for our VE estimates. There was a high probability that contacts had the same vaccination status as the primary case, if they were around the same age (Fig. S7). The probability decreased with the age difference. We found that generally the SAR was highest when both the primary case and contact were unvaccinated. Furthermore, we found a substantial transmission from unvaccinated child primary cases (<20 years) to fully vaccinated adult contacts with a SAR of 24% (Fig. S6)

We found that the probability that secondary cases were infected with the same sublineage of Delta as the primary cases was 88% (95%-CI: 87–89) (Table S3). We found no difference in intra-household correlation across any combination of the vaccination status of primary cases and contacts, indicating no differences in the validity of the matching of primary and secondary cases. However, we did find that unvaccinated contacts were 7 percentage points (10%) less likely to be tested after identification of the primary case. This shows that our main VE estimates are biased downwards.

We performed additional robustness analyses of our main VE estimates, which are presented in Supplementary Section S2.6. All our VE estimates were equal to or higher than the primary estimates, when we restricted our sample to only include contacts that were actually tested after exposure (Tables S5 and S7), indicating that our main VE estimates are biased downwards. We found a reduced VE by time since vaccination, i.e., waning immunity. The pooled $VE_S$ decreased from 71% (95%-CI: 69–72) to 32% (95%-CI: 16–45) between time points corresponding to 0–1 months and 7–8 months after vaccination (Table S8). Similarly, the $VE_I$ decreased from 57% (95%-CI: 53–61) to 29% (95%-CI: 14–41).

## Discussion

We used Danish national population data to estimate household transmission of SARS-CoV-2 Delta VOC to and from unvaccinated and fully vaccinated individuals. We found a vaccine effectiveness against susceptibility to infection in household contacts ($VE_S$) of 61% (95%-CI: 59–63) when the primary case was unvaccinated, and a $VE_S$ of 46% (95%-CI: 40–52) when they were fully vaccinated. Furthermore, we found a vaccine effectiveness against infectiousness from primary cases ($VE_I$) of 31% (95%-CI: 26–36) when the household contact were unvaccinated, and 10% (95%-CI: 0–18) when they were fully vaccinated. Lastly,

**Table 1 Summary statistics.**

| | Fully vaccinated | | | | Unvaccinated | | | |
|---|---|---|---|---|---|---|---|---|
| | Primary Cases | Household Contacts | Secondary Cases | SAR (%) | Primary Cases | Household Contacts | Secondary Cases | SAR (%) |
| **Total** | 8262 | 26,098 | 3816 | 15 | 16,431 | 27,486 | 7815 | 28 |
| **Sex** | | | | | | | | |
| Male | 4001 | 12,868 | 1711 | 13 | 8301 | 13,801 | 3626 | 26 |
| Female | 4261 | 13,230 | 2105 | 16 | 8130 | 13,685 | 4189 | 31 |
| **Age** | | | | | | | | |
| 0–9 years | 0 | 0 | 0 | – | 3668 | 11,687 | 3198 | 27 |
| 10–19 years | 721 | 3482 | 181 | 5 | 4859 | 5742 | 1720 | 30 |
| 20–29 years | 1368 | 2633 | 224 | 9 | 4131 | 3756 | 983 | 26 |
| 30–39 years | 1131 | 4434 | 730 | 16 | 2316 | 3014 | 1028 | 34 |
| 40–49 years | 1676 | 7961 | 1210 | 15 | 923 | 2015 | 585 | 29 |
| 50–59 years | 1601 | 4808 | 686 | 14 | 407 | 994 | 247 | 25 |
| 60–69 years | 1085 | 1893 | 445 | 24 | 109 | 211 | 42 | 20 |
| 70–79 years | 680 | 887 | 340 | 38 | 18 | 67 | 12 | 18 |
| **Household Size** | | | | | | | | |
| 2 persons | 3859 | 4832 | 1209 | 25 | 3704 | 2731 | 812 | 30 |
| 3 persons | 1809 | 5660 | 702 | 12 | 4092 | 5132 | 1402 | 27 |
| 4 persons | 1785 | 9246 | 1204 | 13 | 5080 | 9416 | 2801 | 30 |
| 5 persons | 643 | 4980 | 572 | 11 | 2762 | 7260 | 2029 | 28 |
| 6 persons | 166 | 1380 | 129 | 9 | 793 | 2947 | 771 | 26 |
| **Vaccination** | | | | | | | | |
| Vaxzevria | 514 | 1398 | 229 | 16 | – | – | – | – |
| Janssen | 529 | 546 | 108 | 20 | – | – | – | – |
| Spikevax | 366 | 2441 | 243 | 10 | – | – | – | – |
| Comirnaty | 6853 | 21,713 | 3236 | 15 | – | – | – | – |

The secondary attack rate (SAR) is expressed in percentages. Primary cases and household contacts are here shown by groups of sex, age, household size and vaccination status, independent of each other. Table S1 provides summary statistics for contacts and secondary cases are grouped based on the primary case characteristics.

**Table 2 Vaccine effectiveness (%).**

| | Susceptibility | | | Infectiousness | | | Total |
|---|---|---|---|---|---|---|---|
| Primary cases vaccinated | Pool | Not | Fully | Pool | Not | Fully | |
| Household contacts vaccinated | | | | | | | |
| Estimator | $VE_{S(\cdot, V/\cdot, N)}$ | $VE_{S(N, V/N, N)}$ | $VE_{S(V, V/V, N)}$ | $VE_{I(V, \cdot/N, \cdot)}$ | $VE_{I(V, N/N, N)}$ | $VE_{I(V, V/N, V)}$ | $VE_{T(V, V/N, N)}$ |
| VE (%) | 61 | 61 | 46 | 42 | 31 | 10 | 66 |
| (95%-CI) | (59;63) | (59;63) | (40;52) | (39;45) | (26;36) | (0;18) | (63;68) |
| Age FE | YES | YES | YES | YES | YES | YES | YES |
| Week FE | YES | YES | YES | YES | YES | YES | YES |
| Female | YES | YES | YES | YES | YES | YES | YES |
| Household size FE | YES | YES | YES | YES | YES | YES | YES |
| Female, primary case | YES | YES | YES | YES | YES | YES | YES |
| Age, primary case, FE | YES | YES | YES | YES | YES | YES | YES |
| Ct value, primary case, FE | NO | NO | NO | NO | NO | NO | NO |
| Conditional on test | NO | NO | NO | NO | NO | NO | NO |
| N Observations | 53,584 | 38,336 | 15,248 | 53,584 | 27,486 | 26,098 | 33,822 |
| N Households | 24,693 | 16,431 | 8262 | 24,693 | 15,559 | 16,493 | 20,175 |

This table provides estimates of vaccine effectiveness (%) against susceptibility ($VE_S$) as a pooled estimate ("Pool") as well as stratified by whether the primary case was unvaccinated ("Not") or fully vaccinated ("Fully"). The estimates of vaccine effectiveness against infectiousness ($VE_I$) is given as a pooled estimate and stratified by the vaccination status of the contacts within the household. The total vaccine effectiveness ($VE_T$) is defined as both the primary case and contacts being vaccinated relative to them both being unvaccinated. Note that the VE estimates across columns are not directly comparable as they are estimated on stratified samples. 95% confidence intervals clustered on the household level in parentheses. FE = included as fixed effects in the model. VE estimates conditional on the contacts being tested is presented in Table S5. VE estimates controlling for Ct value of the primary case sample is presented in Tables S6 and S7. VE estimates of waning immunity is presented in Tables S8 and S9.

we found a total vaccine effectiveness ($VE_T$) of 66% (95%-CI: 63–68) when both the primary case and household contact were fully vaccinated compared to both being unvaccinated. Overall, the findings indicate that vaccines are effective in both reducing the susceptibility to infection in exposed contacts and the infectiousness in vaccinated individuals with a breakthrough infection.

Our results show that estimating $VE_S$ and $VE_I$ include several challenges. When few individuals are vaccinated in a population,

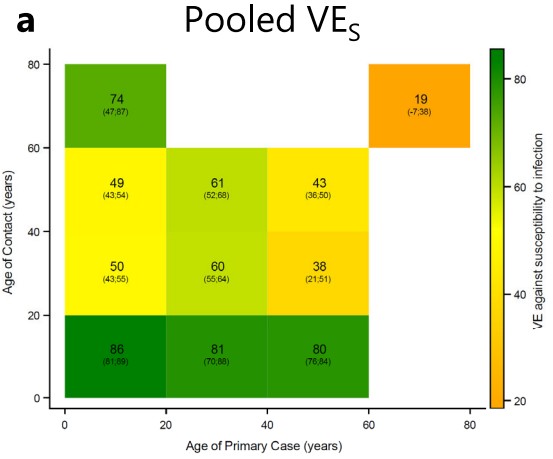
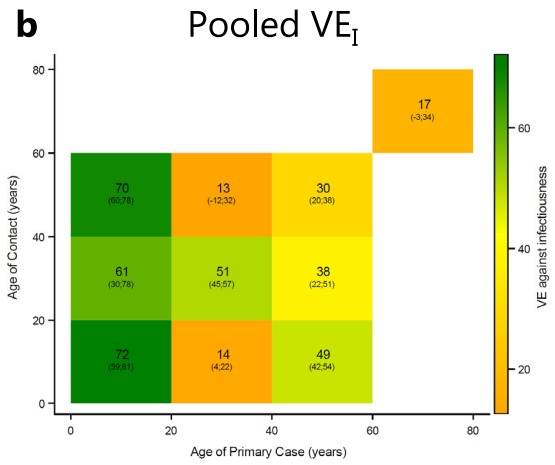

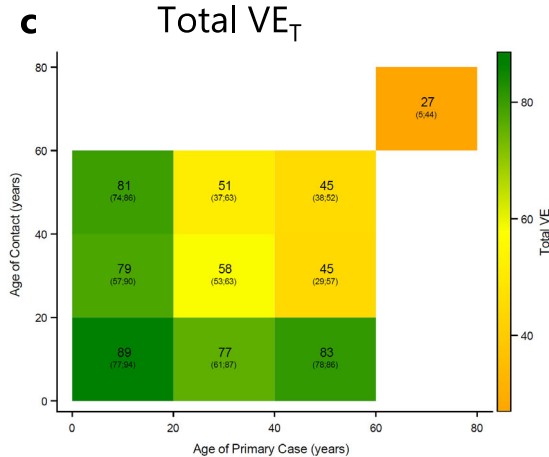

**Fig. 1 Crude VE estimates (%), stratified by age of the primary case and contact.** This figure shows crude VE estimates stratified by age of the primary case and contacts. **a** Shows the pooled $VE_S$. **b** Shows the pooled $VE_I$. **c** Shows the total effect, $VE_T$. 95% confidence intervals clustered on the household level in parentheses. SAR stratified by age and vaccination status of both primary cases and contacts are presented in Fig. S6.

it is easier to estimate the approximate $VE_S$ in exposed contacts because their exposure can be assumed to come from unvaccinated primary cases. As vaccinations are rolled out in a country, the proportion of the population that is vaccinated increases. As a consequence, the proportion of contacts with vaccinated individuals also increases. Therefore, the real-life observed $VE_S$ estimates are a composition of exposure from both vaccinated and unvaccinated primary cases. If vaccination not only protects the exposed contact against infection, but also against infectiousness from the primary case (as we show), then the $VE_S$ estimates are a combined effect of both $VE_S$ and $VE_I$. The $VE_I$ becomes increasingly more important as vaccination rates throughout society increase. The same argument holds for $VE_I$ estimates. This implies that it is necessary to link primary cases to exposed contacts in order to estimate $VE_S$ against infection and $VE_I$, which is difficult in many settings because exposed contacts may not know that they have been exposed or to whom.

Other studies have estimated the $VE_I$ and $VE_S$ with large variation in the results. In a study by de Gier et al.[8], they estimated the $VE_I$ from unvaccinated and vaccinated index cases to 63% and 40%, respectively. A study by Harris et al.[9] found that the $VE_I$ from vaccinated primary cases was 52–54% of that from unvaccinated primary cases. Likewise, Singanayagam et al.[10] estimated the $VE_I$ to 34% and Jalali et al.[11] estimated the $VE_I$ to 42%. All

these studies did not estimate the $VE_S$. A study by Ng et al.[12] found an adjusted $VE_S$ of 61.6% but did not estimate the $VE_I$. Few studies have estimated both the $VE_S$ and $VE_I$ for SARS-CoV-2 Delta VOC from the same data. Prunas et al.[13] estimated the $VE_I$ to 23.0% and the $VE_S$ to 89.4% in Israel. Similarly, Clifford et al.[14] estimated the household $VE_I$ to 14–24% and the $VE_S$ to 31–42% in the UK.

Currently, there is no consensus on the mechanism by which vaccination may affect infectiousness. Immunological theory predicts that vaccinations can inflect an immune response in vaccinated individuals reducing the severity of breakthrough infections, including the viral load, making vaccinated individuals with a breakthrough infection less infectious compared to unvaccinated individuals with an infection. Empirically, however, the effect of vaccination on the viral load is still being investigated. Moreover, the viral load changes over the course of an infection. Thus, it is important to control for the time since exposure, when investigating the differences in the sample viral loads. It has previously been found that vaccinations did not reduce the viral load of cases with a breakthrough infection. Chia et al.[15], for example, found that vaccinated and unvaccinated individuals infected with the Delta VOC had similar Ct values at diagnosis, but that the viral loads decreased faster in vaccinated individuals. Contrary to this, Levine-Tiefenbrun et al.[16] found

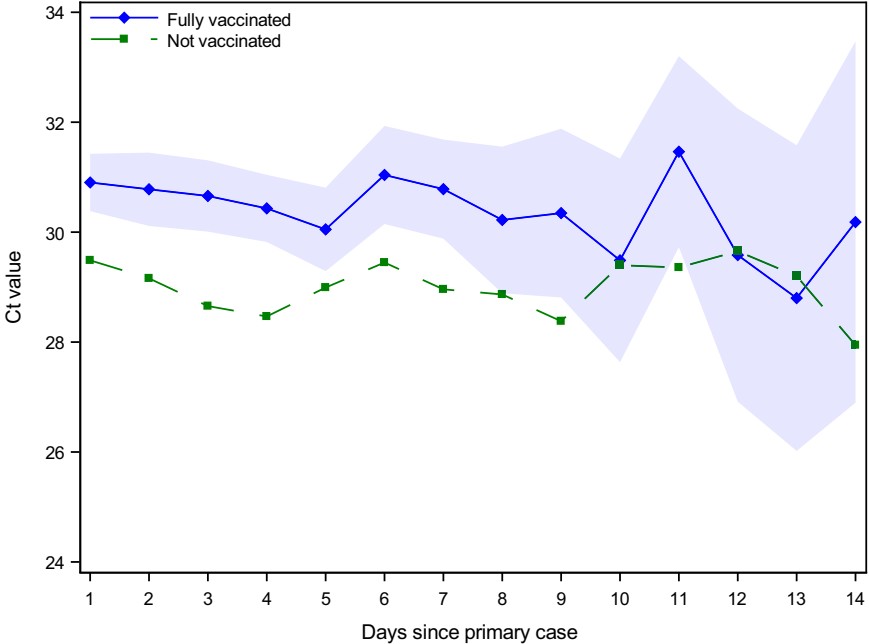

**Fig. 2 Ct values across vaccinated and unvaccinated positive secondary cases.** The Ct values in samples for positive secondary cases of unvaccinated and fully vaccinated follow the same pattern, indicating that the higher Ct values for fully vaccinated individuals is consistent and unrelated to the time of testing positive. Markers present point estimates, while shaded areas are 95%-confidence intervals clustered on the household level. Regression estimates include age fixed effects. Table S4 provides regression estimates of the increased Ct values for vaccinated secondary cases.

that breakthrough infections with the Delta VOC resulted in three times lower viral loads in vaccinated cases compared with unvaccinated cases. A later study found this to be true also for booster vaccination[17]. Our findings corroborate this, as we found that vaccinated secondary cases with a breakthrough infection had lower viral loads (higher Ct values) compared to unvaccinated secondary cases, while controlling for the time since exposure.

Our results have several policy implications. First, our results show that vaccinations can be used to both reduce the susceptibility to infection in exposed individuals and infectiousness in infected cases. This indicates that for pandemic control, it is important to not only prioritize groups that are vulnerable to infection, e.g., nursing home residents, older people, and immune-compromised individuals, but also groups that have many contacts or work with vulnerable individuals, e.g., nursing home staff. This may be increasingly important when deciding on when and whom should be booster vaccinated, as we found evidence of waning immunity in both the estimates of $VE_S$ and $VE_I$. Second, we found that vaccinations protect more against susceptibility to infection than against infectiousness. Therefore non-pharmaceutical interventions (NPIs) acting mostly on the infectiousness, such as wearing masks[18], can be necessary even for vaccinated individuals, especially when they are likely to come into contact with unvaccinated individuals. On the other hand, the reduced susceptibility of vaccinated individuals may facilitate the relaxation of guidelines among vaccinated individuals, e.g., at mass gatherings. This suggests that immunity passports can be an effective measure to reduce transmission, as previously suggested[19]. Third, simulation models have been widely used to inform policymakers about the SARS-CoV-2 pandemic. These models rely crucially on the parameters of susceptibility and infectiousness. Hence, accurate estimates of these effects are critical to the models. Fourth, we here found a substantial degree of transmission to and from children. Unvaccinated children aged 0–10 years were susceptible to the SARS-CoV-2 Delta VOC with a SAR of 27% and unvaccinated primary cases aged 0–20 years

were able to infect vaccinated household contacts aged 20–60 with a SAR of 15%. This implies that children are a key part of SARS-CoV-2 transmission patterns and should not be neglected in pandemic control.

Our approach has several strengths. First, we combined several national data sets allowing us to link primary cases to their household contacts, which allowed us to control for individual specific factors. Second, throughout the study period, Denmark had a large testing capacity that was free and widely used. Third, all household members were per definition close contacts and recommended to become RT-PCR tested twice after the identification of the primary case—unconditional on the vaccination status. Fourth, Denmark had a high vaccination uptake across all vaccination groups. The Danish vaccination program was rolled out to all individuals above 12 years of age during the study period, so unvaccinated individuals mainly represent those who had not yet been invited for vaccination at the time. This is a major strength of our study because a self-selected unvaccinated group might introduce a bias due to the fact that reluctance to receive vaccinations may correlate with other types of behavior. Overall, this provides a setting that allows us to estimate both $VE_S$ and $VE_I$. Furthermore, the rich nature of the data available to us allowed us to explore other ways of verifying the validity of our model, e.g., by looking at the probability that exposed contacts were tested, which is naturally a condition for testing positive. Lastly, during our study period, whole genome sequencing (WGS) of all positive RT-PCR tests were part of the public strategy to control the epidemic. In the present study, the Delta VOC comprised more than 95% of all cases in society and all primary cases were—by definition—infected with the Delta VOC. This made it impossible to investigate variation in the probability that secondary cases were infected with the same variant as the primary case. However, using the subtype lineages of the Delta VOC, we found an overall intra-household correlation of 88% (95%-CI: 87–89%) and no significant differences across vaccinated and unvaccinated primary and secondary cases. Lyngse et al.[20] also used data from WGS to validate the same methods as

used in the present study and found that 96–99% of secondary cases were infected with the same variant as the primary case.

Some limitations apply to this study. Firstly, we did not have access to clinical information, e.g., on symptoms. If vaccinated cases have fewer symptoms than unvaccinated cases, this may lead to unvaccinated cases being identified earlier in their infection than vaccinated individuals. Moreover, due to the rules of the Danish corona passport, unvaccinated individuals may be tested more regularly, which also could lead to them be identified earlier in their infection. Secondly, vaccination status may affect the behavior of both primary cases and contacts. Thus, the estimates in our study reflect both the biological aspect of susceptibility and infectiousness, as well as the behavioral aspects. Once a primary case is identified within the household, the other members might allocate their time with that individual, conditional on their own vaccination status. For example, if the primary case is a child and one parent is fully vaccinated, while the other is not, then the family may choose to allocate the majority of childcare during the infection to the vaccinated parent, as they have a lower risk of being infected. This changed contact pattern may lead to a higher rate of infections in vaccinated individuals in households where there are also unvaccinated members. Furthermore, vaccinated individuals may in general adhere less to NPIs, such as keeping distance, wearing masks, etc. due to a perceived lower transmissibility. However, survey evidence from Denmark show that unvaccinated individuals are less likely to adhere to government recommendations[21]. Lastly, we tested the difference of the probability of being tested after exposure between fully vaccinated and unvaccinated household contacts. We found that vaccinated contacts were about 7 percentage points (10%) more likely to be tested compared to unvaccinated contacts. This suggests that there are differences across the two groups that we cannot fully control for, e.g., general compliance to NPIs. This also implies that our main VE estimates are a lower bound. When we restrict our analyses to only include individuals with a test result, we obtain higher VE estimates.

There is also likely to be a correlation of vaccination status between household members. (i) Individuals living together may be more likely to share the same belief, for instance towards vaccination. (ii) There may be a fixed cost of being vaccinated, e.g., the travel from the home to the vaccination location. Thus, a household might pool their day of vaccination together to minimize travel costs. (iii) Individuals are likely to have a partner around their own age. As vaccination roll-out is based on age, household members are likely to be eligible for vaccination around the same calendar time. Indeed, we found an intra-household correlation of vaccination status of 0.72 for individuals above age 12 years.

Age is correlated with susceptibility and infectiousness as well as eligibility and roll-out of vaccinations. This implies a natural imbalance in the number of primary cases and contacts across age groups. To address this, we provided estimates of SAR and VE in all combinations of age groups, stratified by vaccination status of both the primary case and contact.

The estimates of $VE_S$ are probably conservative compared with the general $VE_S$ at the overall population level. Transmission within household is associated with more intense exposure than in the community in general, and since there is a relation between the degree of exposure and likelihood of breakthrough infection, it is expected that vaccines may work better in the community than in households. Secondly, the estimates based on an infection in a fully vaccinated primary case is naturally conditioned on a breakthrough infection. The virus variant that has caused this infection may have been adapted to vaccine derived immunity, and may therefore be more likely to result in another breakthrough infection in a contact.

From the stratified VE estimates, we find that both the $VE_S > 0$ and $VE_I > 0$. Furthermore, we find a high correlation of vaccination status within households. Combined, this suggests that the pooled VE estimates are confounded, e.g., the pooled $VE_I$ estimate is larger than both of the two stratified $VE_I$ estimates, as it is confounded by the $VE_S$. This underscores the importance of including the stratified analysis in our study, and strongly suggests that analyses of other data sets with similar characteristics should also consider this effect when planning the analysis method.

In conclusion, we have demonstrated that vaccines are effective in reducing transmission of the Delta VOC in Danish households June to October 2021. In particular, we found that vaccines are effective in reducing both the susceptibility to infection in household contacts and the infectiousness of primary cases.

## Methods

**Data**. This study was conducted using Danish register data. In Denmark, all citizens have a personal identification number that allows their information to be linked across different data registers at the individual level. We linked all members of the same household via their registered home address and merged this with data on all RT-PCR tests for SARS-CoV-2 from the Danish Microbiology Database (MiBa) and all positive RT-PCR tests with whole genome sequencing (WGS) in order to identify the variant and sublineage. We then identified the first positive test result in each household and defined the corresponding individual as the primary case. All other registered household members were defined as household contacts. We only considered households with 2–6 members in order to exclude, e.g., student dorms, social housing, and care facilities. We only included households where the primary case tested positive with the Delta VOC (identified by WGS) and where no other household member tested positive on the same day, i.e., excluding co-primary cases. Within each household, we followed household contacts within 0–14 days of the identification of the primary case and define secondary cases as those testing positive within 1–14 days.

*Vaccines*. Information on vaccinations for each individual was obtained from the Danish Vaccination Register (DDV). We classified individuals according to their vaccination status on the test-positive day of the primary case. Individuals who had not received a first dose were classified as not vaccinated. Following the European Medicines agency[22], the definitions of full vaccinations were: Comirnaty (Pfizer/BioNTech): 7 days after second dose; Vaxzevria (AstraZeneca): 15 days after second dose; Spikevax (Moderna): 14 days after second dose; COVID-19 VACCINE Janssen (Johnson & Johnson): 14 days after vaccination. If an individual was cross vaccinated (mainly first dose of Vaxzevria and second dose of Comirnaty), the definition of the second dose vaccination was used. Individuals that were in the period between the first dose and fully vaccinated were defined as partially vaccinated and excluded. Individuals that had received a booster vaccination were also excluded. Lastly, all households with a previous infection (positive RT-PCR test) were excluded.

*Testing*. In Denmark, testing for SARS-CoV-2 was free and widely available during our study period. All individuals living within the same household as a positive case were defined as close contacts and recommended to be RT-PCR tested twice after exposure. As part of the public pandemic response, all positive RT-PCR samples were selected for whole genome sequencing (WGS). See Supplementary Section S1.1 for further elaboration on the testing program.

*Study period*. We selected the study period to include primary cases with the Delta VOC, which became dominant in Denmark around the middle of July 2021. We used data where the primary case within each household was between 21 June 2021 to 26 October 2021. Household contacts were followed up to 9 November to provide sufficient time for them to subsequently test positive. Supplementary Section S1 provides background information on the study period, including the number of tests performed, the number of cases, and vaccination roll-out.

**Statistical analyses**. We defined the overall household secondary attack rate (SAR) as the proportion of household contacts that tested positive between 1-14 days following the identification of the primary case within the same household. We estimated the relative risk (RR) of the SAR of vaccinated individuals compared to unvaccinated individuals, and calculated the vaccine effectiveness (VE) as one minus the relative risk, following Halloran et al.[23]. To estimate the vaccine effectiveness, we used a generalized linear model (GLM), with Poisson distribution response and a log link function, which was fit using maximum likelihood in SAS. The use of a Poisson distribution to describe a binary response was to facilitate estimation of relative risks rather than odds ratios. Standard errors were clustered on the household level. The regression model included fixed effects controls for age (categorical effects in 10-year age groups) and sex of both the primary case and

contacts, and fixed effects for household size (categorical effects). We also included calendar week fixed effects (categorical effects) to control for temporal variation, e.g., behavior, changes in restrictions, vaccination coverage, and overall incidence.

To estimate the extent to which vaccination reduces susceptibility to infection of exposed household contacts, we estimated the relative risk (RR) of the SAR for contacts that were fully vaccinated compared to the SAR for contacts that were not vaccinated. To separate the effect of vaccination affecting susceptibility from the effect on infectiousness, we also stratified by vaccination status of the primary case. In particular, the estimates were defined as $VE_{S(V, V/V, N)}$, when the primary case was fully vaccinated, and $VE_{S(N, V/N, N)}$, when the primary cases were not vaccinated. The pooled estimate—unconditional of the vaccination status of the primary case—was defined as $VE_{S(\cdot, V/\cdot, N)}$.

To estimate the extent to which vaccination reduces infectiousness of the primary case, we estimated the RR of the SAR from primary cases that were fully vaccinated compared to the SAR from primary cases that were not vaccinated. To separate the effect of vaccination affecting infectiousness from the effect on susceptibility, we also stratified by vaccination status of the exposed contact. In particular, the estimates were defined as $VE_{I(V, V/N, V)}$, when the contacts were fully vaccinated, and $VE_{I(V, N/N, N)}$, when the contacts were not vaccinated. The pooled estimate—unconditional of the vaccination status of the contact case—was defined as $VE_{I(V, \cdot/N, \cdot)}$.

To estimate the total effect of vaccination against both susceptibility of the exposed contact and infectiousness of the primary case, we estimated the RR of the SAR for when both the primary case and contact were fully vaccinated compared to when both were not vaccinated, $VE_{T(V, V/N, N)}$.

To investigate the age related transmission patterns, we estimated the $VE_S$, $VE_I$ and $VE_T$ stratified by the age of primary case and contacts.

To explore the effect of vaccinations on reducing the viral load in infected vaccinated cases, we investigated the difference in viral load (represented by the proxy measurement of Ct value) for vaccinated and unvaccinated secondary household cases testing positive on the same day after exposure to the primary case.

A more detailed description of the statistical methods is provided in Supplementary Section S3.

*Additional analyses.* We also performed a number of supplementary analyses to support the main analysis. To investigate if secondary cases were likely to be due to household or community transmission, we investigated the probability that household secondary cases were infected with the same subtype lineage of the Delta variant as the primary case. We investigated the possible bias arising from intra-household correlation of vaccination status, as the vaccination status between household members are likely correlated. Because a positive test result is provided conditionally on actually having a test, we investigated the probability of being tested across vaccination status of both the primary case and contacts. To investigate the robustness of our main results, we estimated the VE conditional on the household contact actually having obtained a test. As age is correlated with time of vaccination and transmissibility, waning immunity is a potential concern. Thus, to investigate the sensitivity of our main results, we estimated the VE by time intervals since vaccination. We also investigated the sample Ct values from unvaccinated and fully vaccinated primary cases. Finally, we controlled for the primary case sample Ct value to investigate if our main results were affected by a change in the viral load of the primary case. The results of the additional analyses are presented in Supplementary Section S2.

**Ethical statement.** This study was conducted on administrative register data. According to Danish law, ethics approval is not needed for this type of research. All data management and analyses were carried out on the Danish Health Data Authority's restricted research servers with project number FSEID-00004942. The publication only contains aggregated results and no personal data.

**Reporting summary.** Further information on research design is available in the Nature Research Reporting Summary linked to this article.

## Data availability

The data used in this study are available under restricted access due to Danish data protection legislation. The data are available for research upon reasonable request to The Danish Health Data Authority and Statens Serum Institut and within the framework of the Danish data protection legislation and any required permission from Authorities. We performed no data collection and performed no sequencing specifically for this study. We used the following Danish administrative registers: Central Person Register (CPR), Danish Microbiology Database (MiBa), Danish Vaccination Register (DDV), and data set on variants for positive RT-PCR tests.

## Code availability

The code used for this study can be downloaded from a public repository: https://github.com/Flyngse/SARS-CoV-2_Delta_Vaccinations.

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

## Acknowledgements

We thank Statens Serum Institut and The Danish Health Data Authority for data access. We also thank the rest of the Expert Group for Mathematical Modelling of COVID-19 at Statens Serum Institut for helpful discussions, as well as the Danish Covid-19 Genome Consortium for typing of positive COVID-19 samples. Frederik Plesner Lyngse gratefully

acknowledge funding from: Independent Research Fund Denmark (Grant no. 9061-00035B.); Novo Nordisk Foundation (grant no. NNF17OC0026542); the Danish National Research Foundation through its grant (DNRF-134) to the Center for Economic Behavior and Inequality (CEBI) at the University of Copenhagen.

## Author contributions

F.P.L. performed data analysis and produced all estimates and figures presented. F.P.L., C.T.K. and K.M. wrote the first draft. M.D. contributed to the selection and description of statistical methods. L.C., C.M., M.R., A.S.C., M.S., J.F., R.N.S., K.M.E., and C.N. contributed to the discussion and writing the final draft. All authors read and approved the final version of the manuscript.

## Competing interests

The authors declare no competing interests.
