## [Peer Review File · Nature Communications]

Effect of Vaccination on Household Transmission of SARS-CoV-2 Delta Variant of ConcernREVIEWER COMMENTS

Reviewer #1 (Remarks to the Author):

This is a very relevant study using a large and valid dataset to estimate vaccine effectiveness of susceptibility (as the authors phrase it) and of transmissibility during the period where the Delta variant was dominant. Although outdated (as everything quickly is with COVID-19), the study should be published because of its good design and elaborate analyses which can be an example for future studies on this topic.

I have several comments or questions, which should be clarified by the authors.

1. Could the authors explain why they have chosen the term VE of susceptibility (VES). I would see this as a VE against infection in a population who has been in contact with a positive case.
2. Why was 21 June chosen as the start date, while the Delta variant was only dominant in July? Related to this, how can you be sure that the primary case was indeed infected with the Delta variant if also other variant were still circulating?
3. Why were different periods after vaccination chosen for different vaccines to define someone as fully vaccinated, e.g. 7 days for Com, 14 days for Spikevax and 15 days for Vax. You could argue about the correct time interval (7 or 14 days) but to do this differently for different vaccines is unusual. So this should be explained. Related to this, were there difference in VE estimates between vaccines?
4. It is not entirely clear how calendar week was included in the models. As a categorical variable? The authors could consider to use a spline to adjust for calendar week.
5. An additional analysis that the authors describe is adjusting for viral load. It was not clear to me what the aim was of this analysis. To assess whether viral load is a mediator/intermediate in the causal pathway from vaccination to transmission?
6. In the VE analysis the authors stratify for vaccination status of the primary/secondary case. For VET they find an overall estimate of 42%, and 31% for unvaccinated contacts and 10% for vaccinated contacts. This means that VET is confounded by vaccination status of the contact because the weighted average of VET (I would call this the pooled VET) over unvaccinated and vaccinated contacts will be lower than the overall VET of 42%. So that makes the overall VET of 42% invalid, I would say. It is also confusing the authors call this the pooled VET as for me that would mean the weighted average of VET over unvaccinated and vaccinated contacts.
7. The authors perform a lot of additional and sensitivity analyses, which are very informative. These are not presented in the results section which I think is a missed opportunity. The discussion on the other hand is very long and includes many (new) results. At least the results on waning immunity and the results stratified by age group should be presented in the main results rather than in the appendix. Also the different sensitivity analyses could be summarized in a (forest plot-like) figure in the main results as this would make the robustness of the results over different exclusions etc. much more clear.
8. The discussion could be improved by more comparison with findings in literature. There is not that much literature on VET, especially not for Delta. But for VE against infection, which VES and VEC are in my view, there are many results from other studies.
9. I do not completely understand the paragraph in the discussion on the difference between VEC, VES and VET. I am not convinced that VEC is a combination of VES and VET. I think VEC is 'just' a VE against infection and VES is also a VE against infection but within persons with known (and intense) exposure to SARS-CoV-2. The reason for VEC to be higher than VES is probably the certain but also more intense exposure.
10. Figure 2 in the Appendix shows that the percentage of positive tests increased over the study

period. Is this a result of availability of self-administered tests with the consequence that persons testing negative at home don't seek testing and that persons testing positive at home seek confirmation testing? If so, will that have influenced your results?

11. Was the quality/extensiveness of source and contact tracing constant over the study period?

12. Figure 8 of the appendix deserves to be in the main part of the manuscript. The authors chose dark green to indicate a lower VE, which seems counterintuitive as green is generally used for something good. Please consider to adapt this.

13. Table 10 shows a lot of waning of immunity over time since vaccination, more than in other studies I believe. When conditioning on having a test (Table 11) waning is less severe. I think this deserves some discussion and also comparison with other studies.

Reviewer #2 (Remarks to the Author):

VE on transmission after breakthrough infection is a relevant topic and as the paper states, more data is needed on this topic. In general, this paper can help answer this question. The paper benefits from extensive testing and (good?) household data. My two main issues with the paper are the selection of the data and the way in which the analysis is performed. Given the 'remarkable' results: VET is highest when all data is analyzed, larger than when the data is stratified into either non/vaccinated index cases, I am hesitant to "trust" the paper. In addition, there are a number of inconsistencies (e.g. table 3, no vaccinated contacts of child-index cases?, figure 10 including data from day 0, which should be excluded?).

Out of habit, I will often refer to the primary case as 'index' and to the household member, potential secondary case, as 'contact'.

Data included:

Why is the 'test' data only included as sensitivity analysis and not as main analysis:

Household members are considered observations in the paper, but we do not actually know if they have been observed. The study period spans the summer holidays. Is it not plausible that some household members were not at the household? (E.g. kids on summer camps?) It seems like only around 80% of potential cases were tested (figure 10). Lower if this percentage really includes those tested on day 0 (which are excluded?). The authors include the "test only" data as sensitivity analysis, but it is unclear why this was not the main analysis?

What data is included:

It is also unclear how exactly the two (main analysis and sensitivity analysis) correspond to one another. Who is tested and who is not tested?

-In one of the additional analysis it is shown that unvaccinated persons were less frequently tested, this is likely confounded (at least in some part) by the age of the household member (young kids being unvaccinated and tested less? (line 566)). It seems essential info, but it is not possible to obtain it from the paper/appendix.

-I would be very interested on the proportion tested the day after the index was tested (as for these persons another source of infection, perhaps common to that of the index can be suspected). From figure 10, it seems like a large proportion of contacts were tested on day 0. Figure 10: this figure runs from 0 to 14 days after primary case, while in the methods (line 83) it says that only 1:14 days are included. This has a potentially large effect, as from figure 10 it seems like around 30% have their test on day 0. Other figures run from day 1 to 14.

More information on attrition is necessary.

I am surprised by the high number of primary cases 0-9 years of age (especially given the summer period). I know case ascertainment is not the major focus of this paper, but I presume that case finding was not limited to symptomatic infection?

Analysis:

While it is a simple analysis (GLM) I cannot follow the different 'stratifications'/subsets and, to me, it is unclear which models were fitted to which data. The authors work with a 'pooled' estimate, a 'not' and a 'fully'. As I understand, these are all fitted to different subsets/stratifications of the data. I would assume that the pooled is fitted to all available data, but then it is very hard for me to understand how the pooled estimate can be larger than both the 'not' and 'fully' estimate. It also makes the interpretation of the analysis hard (VET is highest when 'pooled', higher than when the contact is unvaccinated or vaccinated. So in which scenario is VET 41%?). Why are the vaccination status of contact and index not included in the same model so the VET-fully and VET-not can be estimated from the same dataset?

As the authors indicate in the introduction, waning is complicating VE-estimation. While results from a 'time since vaccination' analysis are reported in the paper (analysis in appendix), it is unclear why this analysis is given less weight than the not/fully/pooled analysis as waning (from the sensitivity analysis) seems to have a very large impact on the results. E.g. are the non-significant results for the VES and VET in persons aged 60-80 a result of waning?

The authors are not consequent with the V,N notation. ('.' should represent pooled, but it is not used that way in the tables in the paper).

Table 3 (characteristics stratified by primary case) seems wrong. Were there no primary cases aged 0-10 that had fully vaccinated potentially secondary cases?

The Ct-analysis is also not clear to me, are the I, II, III in the heading representing different regression models? Which method was used to conclude on a significant difference, while it being one third of the standard deviation...

In table 10, there is a sign negative VE, is there any hypothesis for this finding?

Some smaller remarks:

Abstract

*Total population data (I think I understand it from the paper (total = all Danish cases during the study period and their household are considered for inclusion), but from the abstract it is unclear what this means).

*Specify vaccine effectiveness (full schedule/partial vaccination)

*Should the difference in Ct-values be reported in the abstract given only 'one third of sd'

Introduction

35* "difficult to estimate VE because of prioritized vaccination and waning". This is true, but the link ('it is therefore essential') with VES and VET is unclear.

52* "individuals may change their behavior due to the vaccination status of themselves"

Is there any indication that this is true for households?

Methods

Why was this period chosen?

Some sort of attrition table would be nice (loss because of partial vaccination, alpha-infection etc.)

Uncertainty estimation

Was the GLM fitted using ML or Quasi ML?

Different age groups are mentioned:

109: 5-year age groups

660: 10-year age groups

Results

170: rounding error => 51%

Table 1: 1380+2947 potential secondary cases in households of 6, why is this number not a multiple of 5? The elimination of partially vaccinated?

240: Therefor

Best Regards,
Toon Braeye

Reviewer #3 (Remarks to the Author):

This article uses household testing data from Denmark to assess vaccine effectiveness for reducing susceptibility to infection and for reducing infectiousness given infection. These collectively impact transmission potential and are important parameters for mathematical models and guide policy decisions. The data are collected during the Delta era from a large number of households. The data themselves are interesting, but I have some concerns about how they are analyzed and communicated. I was also distracted by the terminology the authors use to describe these effects, and I would strongly prefer they used more commonly accepted names that have been around since the 1990s.

Major comments:

1. Terminology.

- It would be better to use the recognized terminology of Halloran et al., which defines VE_I as vaccine effectiveness for infectiousness given infection instead of VE_T (which the authors label as transmissibility). In the existing literature, VE_T is used to represent total vaccine effectiveness (the combined impact of direct VE and indirect VE accrued to a vaccinated individual).
- VEC (VE_combined) is not a standard measure. From the Halloran and Preziosi 2003 reference, combined refers to an average SAR across groups, not the combined effect of multiple processes. The description in line 99 is not sufficiently clear ("vaccinated individuals compared to unvaccinated individuals" is too vague). Based on their statistical appendix, VEC would be better described as VE_T or total vaccine effectiveness.
- Abstract. Clarify that VE_S is susceptibility to infection by adding in reference to infection.
- Line 41. Transmission here is not clearly defined. (Is it defined unconditionally or conditional upon infection?) Clearer terminology is infectiousness given infection. Alternatively, "from infected vaccinated individuals" and "from infected unvaccinated individuals."
- Line 347. "VE against transmissibility among unvaccinated potential secondary cases" is difficult phrasing. The infectiousness is coming from the index cases, and TO unvaccinated household contacts.

2. Calculation of VE in models.

- Line 173. Define pooled VEs. Does the pooled estimate model for susceptibility include a term for vaccination status of the index case? Or is this handled only through stratification?
- In Table 2, the pooled estimate for VE against infectiousness (42%) is not between the two stratified estimates (31% and 10%) but is instead higher. This makes me think of Simpson's paradox and confounding due to clustering of vaccination within households. It makes me wonder how interpretable the primary estimate is.
- Similarly, why does the pooled VE for susceptibility 61% (59,63%) have an identical point estimate and 95% confidence interval to the stratified not vaccinated group? Can the authors confirm this is not an error?

Minor:

- Abstract should include the vaccines in use in Denmark and the time period of the study.
- Abstract: Meaning of "(one third of a standard deviation)" unclear. "A viral load one third of a standard deviation higher than fully vaccinated individuals"?
- Abstract: "Our results imply that vaccinations reduce susceptibility"... mention the Delta variant.
- Introduction. "SARS-CoV-2 pandemic" and "COVID-19 infections" would be better rewritten as "COVID-19 pandemic" and "SARS-CoV-2 infections", and similar.
- Line 40. Provide context for Harris study. Households in England, time period.
- Line 45. It is unfortunate for this paragraph to not cite any of the broad literature on using household studies to estimate VE_I.

- Line 60. "with and without vaccination against SARS-CoV-2" should be deleted, as VE is itself a relative measure.
- Section 3.2. Description of testing protocols in Denmark in this section would be helpful. They are mentioned later in the discussion, but provide important context on how complete the results might be.
- Line 105. Unclear what "see below" refers to.
- Table 2 includes helpful notation defining the estimators, but this notation is not itself defined in the main text.

Comments to all reviewers

We thank you all for your time and effort in providing detailed and extremely helpful reviews, which we believe have contributed to greatly improve the manuscript.

In accordance with your comments, we have re-written the full manuscript. We have not included track changes, as we believe this would be more of a nuisance than a help due to the large number of small changes.

Please find our point-by-point responses to your remarks below.

Reviewer #1 (Remarks to the Author):

RI *This is a very relevant study using a large and valid dataset to estimate vaccine effectiveness of susceptibility (as the authors phrase it) and of transmissability during the period where the Delta variant was dominant. Although outdated (as everything quickly is with COVID-19), the study should be published because of its good design and elaborate analyses which can be an example for future studies on this topic.*

AU We thank the reviewer for the kind words and agree that the study is of general interest, despite the shift of predominant variant and changing situation throughout the world.

RI *I have several comments or questions, which should be clarified by the authors.*

RI.Q1 *1. Could the authors explain why they have chosen the term VE of susceptibility (VES). I would see this as a VE against infection in a population who has been in contact with a positive case.*

AU We agree. This is VE against infection given exposure. Thus, the analysis on susceptibility relates only to the probability of acquiring infection given exposure. We have made this clearer throughout the manuscript.

RI.Q2 *2. Why was 21 June chosen as the start date, while the Delta variant was only dominant in July? Related to this, how can you be sure that the primary case was indeed infected with the Delta variant if also other variant were still circulating?*

AU We chose 21 June as start date in order to include as many households as possible. At the time of the study, all positive RT-PCR cases were whole genome sequenced (WGS) as part of the Danish covid-19 pandemic surveillance strategy. Thus, we use the WGS result of the primary case sample to identify the variant. We only included households with the Delta VOC. This allows us to include cases from early on, i.e., before Delta became dominant. We have now clarified how we identify the variant in the Methods section.

RI.Q3 3. *Why were different periods after vaccination chosen for different vaccines to define someone as fully vaccinated, e.g. 7 days for Com, 14 days for Spikevax and 15 days for Vax. You could argue about the correct time interval (7 or 14 days) but to do this differently for different vaccines is unusual. So this should be explained. Related to this, were there difference in VE estimates between vaccines?*

AU To have an operational definition of the time when full protection occurs, we follow the official periods, which are available from EMA in the summaries of product characteristics (EPAR) (<https://www.ema.europa.eu/en/human-regulatory/overview/public-health-threats/coronavirus-disease-covid-19/treatments-vaccines/covid-19-vaccines>), see limitations of vaccine effectiveness in section 4.4. The definition of full vaccination differs between the vaccines, and therefore we used varying periods after vaccination. We briefly looked into differences across vaccines, but the majority (83%) of the population having been vaccinated with Pfizer (Table 3). Additionally, the administration of some types of vaccines were correlated with age or occupation. For example Johnson&Johnson were primarily offered to young males and AstraZeneca were primarily given to healthcare workers with a second dose of Pfizer. Therefore, we decided to not push further in this direction.

RI.Q4 4. *It is not entirely clear how calendar week was included in the models. As a categorical variable? The authors could consider to use a spline to adjust for calendar week.*

AU Calendar week was included in the models as a categorical variable, i.e., as a fixed effect. We have now improved the description of that in the methods section. We agree with the reviewer that a spline could be an alternative, but we would not really have benefitted from reducing the dimensionality of the explanatory variable in this way due to the large amount of data available to us. Furthermore, fitting independent fixed effects allows for temporary shifts in behavior due to e.g. national holidays on a week-by-week basis that a spline may smooth out.

***RI.Q5** 5. An additional analysis that the authors describe is adjusting for viral load. It was not clear to me what the aim was of this analysis. To assess whether viral load is a mediator/intermediate in the causal pathway from vaccination to transmission?*

AU Yes. The aim of the analysis adjusting for viral load was to investigate if the estimates of susceptibility and infectiousness was associated with the viral load, as viral load can be affected by the vaccination status of the primary case (Figure 2 [original manuscript: Figure 1]). We have now clarified that in the Methods section.

***RI.Q6** 6. In the VE analysis the authors stratify for vaccination status of the primary/secondary case. For VET they find an overall estimate of 42%, and 31% for unvaccinated contacts and 10% for vaccinated contacts. This means that VET is confounded by vaccination status of the contact because the weighted average of VET (I would call this the pooled VET) over unvaccinated and vaccinated contacts will be lower than the overall VET of 42%. So that makes the overall VET of 42% invalid, I would say. It is also confusing the authors call this the pooled VET as for me that would mean the weighted average of VET over unvaccinated and vaccinated contacts.*

AU We agree with the reviewer. We chose to use the terminology “pooled” estimate, following Halloran (2003). Many studies use the pooled estimate without controlling for the vaccination status of the other part—typically not controlling for the vaccination status of the primary case and only looking at the vaccination status of the contact. This is one of the points that we would like to communicate with our study. We have now made this clearer in the manuscript, however, we choose to keep the terminology from Halloran (2003).

RI.Q7 7. *The authors perform a lot of additional and sensitivity analyses, which are very informative. These are not presented in the results section which I think is a missed opportunity. The discussion on the other hand is very long and includes many (new) results. At least the results on waning immunity and the results stratified by age group should be presented in the main results rather than in the appendix. Also the different sensitivity analyses could be summarized in a (forest plot-like) figure in the main results as this would make the robustness of the results over different exclusions etc. much more clear.*

AU We have now included a brief presentation of the additional analyses in the results section and appendix. Furthermore, we have highlighted the results on waning immunity in the results section, but do not want to promote it as a main result. Because the additional analyses are conducted to check for robustness of the results in the main analysis, we prefer keep the current format of the results.

RI.Q8 8. *The discussion could be improved by more comparison with findings in literature. There is not that much literature on VET, especially not for Delta. But for VE against infection, which VES and VEC are in my view, there are many results from other studies.*

AU We have now updated the discussion to include the latest published results from other studies.

RI.Q9 9. *I do not completely understand the paragraph in the discussion on the difference between VEC, VES and VET. I am not convinced that VEC is a combination of VES and VET. I think VEC is 'just' a VE against infection and VES is also a VE against infection but within persons with known (and intense) exposure to SARS-CoV-2. The reason for VEC to be higher than VES is probably the certain but also more intense exposure.*

AU VE_T [original submission: VEC] is the total observed effect of vaccination with contributions from reduced susceptibility and reduced infectiousness. VE_S is the VE against susceptibility to infection of the exposed contact. VE_I [original submission: VET] describes the effectivity against infectiousness in the primary case. We have now clarified this in the methods section, and also revised the terminology in order to improve the readability.

We believe that all contacts in our study has a known and intense exposure to SARS-CoV-2, as they live in a household with a primary case. However, both the infectiousness of the primary case and the susceptibility of the contact may be affected by vaccination. The combination of both vaccination of the primary case and vaccination of the household contact is reflected in the VE_T [original submission: VEC].

RI.Q10 10. *Figure 2 in the Appendix shows that the percentage of positive tests increased over the study period. Is this a result of availability of self-administered tests with the consequence that persons testing negative at home don't seek testing and that persons testing positive at home seek confirmation testing? If so, will that have influenced your results?*

AU Self-administered tests were not widely available in the study period, but the increase in prevalence of positive tests was caused by a combined seasonal effect and higher spreading after ease of restrictions and people returning to normal life with many physical contacts, e.g., at schools and workplaces. We have now added an explanation of this to the manuscript (Lines 559-560).

RI.Q11 11. *Was the quality/extensiveness of source and contact tracing constant over the study period?*

AU Yes, there were no changes in the contact tracing over the study period. Furthermore, household contacts have been defined as close contacts with an encouragement to be tested throughout the full epidemic in Denmark. It should be relatively easy for the primary case to inform the rest of their household of the exposure and need to be tested. To further investigate this question, we have included Appendix Figure S8b showing the proportion of household contacts being tested after exposure stratified by calendar week, as a proxy for contact tracing. Note, the Danish school summer vacation ran from week 26 to 31 (both weeks included), 2021. This shows that there are differences in the contact pattern during the summer time (including the holidays) and the autumn, as we also discuss in the discussion. However, the analysis controls for calendar time, which can be regarded as a proxy for the changes in contact tracing practices.

RI.Q12 12. *Figure 8 of the appendix deserves to be in the main part of the manuscript. The authors chose dark green to indicate a lower VE, which seems counterintuitive as green is generally used for something good. Please consider to adapt this.*

AU Thank you for this suggestion. We agree that the colors may be counterintuitive, and have switched them around. We have also included this figure in the main manuscript as Figure 1.

RI.Q13 13. *Table 10 shows a lot of waning of immunity over time since vaccination, more than in other studies I believe. When conditioning on having a test (Table 11) waning is less severe. I think this deserves some discussion and also comparison with other studies.*

AU We agree and have now included this in a separate paragraph of the Discussion. However, estimating waning immunity is not an objective of our study: it is included in our manuscript to show how our estimates are sensitive to this potential confounder, and not specifically to provide a precise estimate of waning immunity.

Reviewer #2 (Remarks to the Author):

R2 *VE on transmission after breakthrough infection is a relevant topic and as the paper states, more data is needed on this topic. In general, this paper can help answer this question. The paper benefits from extensive testing and (good?) household data. My two main issues with the paper are the selection of the data and the way in which the analysis is performed. Given the ‘remarkable’ results: VET is highest when all data is analyzed, larger than when the data is stratified into either non/vaccinated index cases, I am hesitant to “trust” the paper.*

AU We are happy that you like the topic and that the manuscript can help answer these pertinent questions. Please see our answers below to the inquiries about the data selection and the analysis.

R2.Q1 *In addition, there are a number of inconsistencies (e.g. table 3, no vaccinated contacts of child-index cases?, figure 10 including data from day 0, which should be excluded?).*

AU Table S1 [original manuscript: Table 3] is stratified by the primary case level. Therefore, it should be read as “there are no contacts or secondary cases for vaccinated children”. This is because the vaccine roll-out did not include children below the age of 12 in the study period.

Figure S8 [original manuscript: Figure 10] shows the probability that household contacts are being tested. This is naturally important for the validity of our results of testing positive, which are affected by the propensity to be tested. Household contacts can be tested on the same day as the sample date of the primary case’s positive test. This is why we include day=0 in this figure. We have now made this clearer in the Methods section.

R2.Q2 *Out of habit, I will often refer to the primary case as ‘index’ and to the household member, potential secondary case, as ‘contact’.*

AU We agree that the potential secondary cases are better described as household contacts, and have corrected this throughout the manuscript. In public health, the primary case is usually defined as the first case of a transmission chain, whereas the index case is the first identified case (often related to a field investigation of an outbreak¹). Using the timing of tests and test results, we identify the primary case. Thus, we believe this is the correct terminology for our setting.

Data included:

R2.Q3 *Why is the ‘test’ data only included as sensitivity analysis and not as main analysis: Household members are considered observations in the paper, but we do not actually know if they have been observed. The study period spans the summer holidays. Is it not plausible that some household members were not at the household? (E.g. kids on summer camps?) It seems like only around 80% of potential cases were tested (figure 10). Lower if this percentage really includes those tested on day 0 (which are excluded?). The authors include the “test only” data as sensitivity analysis, but it is unclear why this was not the main analysis?*

AU We understand the concerns of the reviewer and agree that the potential differential bias introduced by the propensity to test is critical to carefully examine and understand. In general, there will be three reasons for household members to not have been tested: (1) they believe themselves to be uninfected due to, e.g., previous infection and/or lack of symptoms, (2) they have a personal/political belief that reduces their compliance with covid-19 controls, and (3) they are not at risk of infection due to not being in contact with the primary case (due to, e.g., summer camps as stated by the reviewer). Each of these three groups would be expected to introduce bias in different directions: for example excluding (1) from the analysis will bias the transmission rate upwards, and including (3) in the analysis may bias the transmission rate downwards. However, it is not possible to determine the reason for non-testing at individual level, so we must choose the same policy for all non-tested individuals. We therefore chose to include both possible strategies, i.e., include non-tested and exclude

¹ In field epidemiology, the index case is the first identified case in a cluster of case-patients that prompts an outbreak investigation. The index case may be identical to the primary case, but it is also possible that a primary case is identified during the field investigation, i.e., if the index case did not represent the first case in the chain of transmission.

non-tested, so that we obtain estimates for all parameters including bias in both possible extremes. Given that our estimates are consistent, we believe that this is strong evidence that our results are robust to these issues.

Furthermore, a major strength of our analysis is that we are able to investigate and quantify the underlying differences in obtaining a test across vaccination status, which is a natural condition for obtaining a test result. We think that showing the difference in the testing propensity and showing how the conditioning affects our estimates is a transparent way to show the sensitivity of our results. Moreover, more than 70% of household contacts were tested (Appendix Figure S8), which is a relatively high proportion compared to other studies. (See, e.g., <https://academic.oup.com/cid/article/74/3/407/6273394?login=true>, in which they find a SAR of 9% (231,498/2,474,066). However, only 16% (392,737/2,474,066) of the traced contacts were actually tested.

Lastly, summer holidays likely affect the household contact patterns. However, it is not clear in which direction. On one hand, children are not in school and families spend more time together, which increase the within-family contact patterns. On the other hand, households might spend time apart, e.g., the children on summer camp away from their parents. One way to empirically address this question is to investigate the testing propensity of household contacts over time: we have now included Appendix Figure S8.b to do this. Furthermore, at this period in Denmark, intensive testing had to be carried out before engaging in any gatherings, such as summer camps. Therefore, it is less likely that we have included families with kids on summer camps. Additionally, there was a public requirement for isolation after testing positive, and being contact to a positive individual, in the study period. It might also be reasonable to assume that children testing positive while being on summer camp relocate back home to be with their family to be cared for by their parents. For these reasons, we chose to present the analysis including all non-tested individuals in the main paper, with the analysis excluding non-tested individuals in the supplementary material.

What data is included:

R2.Q4 *It is also unclear how exactly the two (main analysis and sensitivity analysis) correspond to one another. Who is tested and who is not tested?*

AU We think that there might be a misunderstanding regarding the “conditional on test” formulation. By that term we mean that the analysis was done in the subpopulation of household contacts that were tested in the follow-up period (see Figure S8). We have now clarified that throughout the manuscript. Please also see our response to R2.Q3.

R2.Q5 *-In one of the additional analysis it is shown that unvaccinated persons were less frequently tested, this is likely confounded (at least in some part) by the age of the household member (young kids being unvaccinated and tested less? (line 566)). It seems essential info, but it is not possible to obtain it from the paper/appendix.*

AU We have now expanded our appendix section on testing propensity and included an extra panel figure in Figure S8 containing information with the testing propensity by age group, Figure S8c.

R2.Q6 *-I would be very interested on the proportion tested the day after the index was tested (as for these persons another source of infection, perhaps common to that of the index can be suspected). From figure 10, it seems like a large proportion of contacts were tested on day 0. Figure 10: this figure runs from 0 to 14 days after primary case, while in the methods (line 83) it says that only 1:14 days are included. This has a potentially large effect, as from figure 10 it seems like around 30% have their test on day 0. Other figures run from day 1 to 14. More information on attrition is necessary.*

AU Figure S8 [original manuscript: Figure 10] shows the probability that household contacts are being tested. This is naturally important for the validity of our results of testing positive, which are affected by the propensity to be tested. Household contacts can be tested on the same day as the sample date of the primary case’s positive test. This is why, we include day=0 in this figure. We have made this clearer in the Methods section now. We have

increased the information in Figure S8.a to include “tested once” and “tested twice”. Furthermore, below, we show the daily point estimates for the probability of obtaining a test.

Response Figure: Point probability of being tested for household contacts relative to the primary case sample date

Notes: This figure presents the point probability for household contacts for being tested relative to the primary case. Shaded areas are 95%-confidence bands clustered on the household level.

We define secondary cases as those testing positive within 1-14 days of the primary case. Contacts testing positive on day zero are considered co-primary cases. All households with co-primary cases are excluded. Figure 2 [original manuscript: Figure 1] shows the Ct values across vaccinated and unvaccinated secondary cases. This is conditional on having a positive test and therefore run from day 1 to 14.

Lastly, the table below shows the overall choices made in selecting the data for the analysis, including the number of contacts and household clusters affected for the selected time period.

Response Table: Number of contacts and households for data analysis

	Number of contacts	Number of households
All WGS lineages	108,211	48,455
only Delta primary cases	82,413	36,862
Include partial vaccinated individuals	81,471	36,326
Include booster-vaccinated individuals	58,018	27,094
Include co-primary case households	57,676	26,924
Final	53,584	24,693

R2.Q7 I am surprised by the high number of primary cases 0-9 years of age (especially given the summer period). I know case ascertainment is not the major focus of this paper, but I presume that case finding was not limited to symptomatic infection?

AU That is correct. Cases were only defined by register data on test results. In Denmark, extensive testing of the general public was carried out at that time, making it possible to identify both symptomatic and asymptomatic cases. Furthermore, in Denmark, household contacts have been defined as close contacts with an encouragement to be tested throughout the full epidemic. In Denmark, families often spend time together during the summer vacation (usually lasts three weeks), whereas other countries have an extensive tradition for summer camps and other organized activities targeting school aged children. Hence, transmission of SARS-CoV-2 to and from children in the household domain is something that can be expected throughout the year.

Analysis:

R2.Q8 *While it is a simple analysis (GLM) I cannot follow the different 'stratifications'/subsets and, to me, it is unclear which models were fitted to which data. The authors work with a 'pooled' estimate, a 'not' and a 'fully'. As I understand, these are all fitted to different subsets/stratifications of the data. I would assume that the pooled is fitted to all available data, but then it is very hard for me to understand how the pooled estimate can be larger than both the 'not' and 'fully' estimate. It also makes the interpretation of the analysis hard (VET is highest when 'pooled', higher than when the contact is unvaccinated or vaccinated. So in which scenario is VET 41%?). Why are the vaccination status of contact and index not included in the same model so the VET-fully and VET-not can be estimated from the same dataset?*

AU We follow the methods from Halloran (2003) and therefore stratify the data. We have now re-written the methods section to make the sample stratification more clear.

In the interpretation, it is important to consider the different stratifications and how vaccines may have an impact in these different scenarios. Thus, the marginal effect of a vaccine is higher when the recipient is exposed to an unvaccinated primary case than a fully vaccinated case. It will be beyond the scope of this paper to speculate about the biology, but it could be related to a selection of "vaccine-resistant" virus strains in this scenario. As an alternative explanation, it may be related to the fact that VE is calculated based on relative risks and not risk differences. Clearly, the lowest attack rates are seen in household with high uptake of vaccinations, and this is not well reflected by the relative measures of association. In other words, the baseline level of infection risk is of importance in the interpretation of the data.

Likewise, the reduced infectiousness rendered by vaccines is more obvious when the household contacts are unvaccinated and become less evident when the household contacts are vaccinated (a marginal effect of 10%).

The pooled estimates are provided across the scenarios and populations, and are more likely to uncover estimates relevant for public health purposes than the stratified analyses in somehow artificial populations.

R2.Q9 *As the authors indicate in the introduction, waning is complicating VE-estimation. While results from a ‘time since vaccination’ analysis are reported in the paper (analysis in appendix), it is unclear why this analysis is given less weight than the not/fully/pooled analysis as waning (from the sensitivity analysis) seems to have a very large impact on the results. E.g. are the non-significant results for the VES and VET in persons aged 60-80 a result of waning?*

AU We agree that this analysis is interesting. However, it is not our main focus. Parameter identification of waning VE is difficult to estimate credibly, because time since vaccination is so closely connected with the age of the individuals due to the vaccine roll-out strategy. It is however important to understand how sensitive our main results are to different specifications of the model, which is why we include these analyses as supplementary analyses. We have now re-written the introduction—and excluded waning immunity to not confuse the reader about the main focus of the paper.

R2.Q10 *The authors are not consequent with the V,N notation. (‘.’ should represent pooled, but it is not used that way in the tables in the paper).*

AU Thank you for this observation. Indeed, the V,N-notation for all tables had been switched around. This is now corrected.

R2.Q11 *Table 3 (characteristics stratified by primary case) seems wrong. Were there no primary cases aged 0-10 that had fully vaccinated potentially secondary cases?*

AU Appendix Table S1 [original manuscript: Table 3] is correct. The table is the same as Table 1, but stratified by the primary case level. There were no primary cases <12 years that were vaccinated. Hence, they also did not have any contacts. Indeed, many primary cases <12 years had fully vaccinated contacts, which is partially represented in Appendix Figure S6 and Table S2.

R2.Q12 *The Ct-analysis is also not clear to me, are the I, II, III in the heading representing different regression models? Which method was used to conclude on a significant difference, while it being one third of the standard deviation...*

AU Yes, the roman numbers refer to the different regression models used in the sensitivity analyses. We have now made this clear throughout the text. We believe model II [original manuscript: model III] is the most correct and have re-written the paragraph to make this clear. The other model specifications are there for transparency.

R2.Q13 *In table 10, there is a sign negative VE, is there any hypothesis for this finding?*

AU We ascribe this to the difficulty in credibly estimating the VE parameters, as time since vaccination is so closely connected with the age of the individuals due to the Danish vaccine roll-out strategy. See also response to R2.Q8. In the table note, we also address this: “Note, the negative VE_I estimates, when the household contacts were fully vaccinated, suggest that there is bias in the comparison of the vaccinated and unvaccinated population that we do not fully control for. We did not find negative VE_I estimates, when we conditioned on the household contacts having a test result (Appendix Table S9), indicating that differences in the probability of being tested across unvaccinated and fully vaccinated contacts is a bias in our model (Appendix S2.4). Other behavioral biases across unvaccinated and fully vaccinated individuals are also likely. Thus it is most likely that there is a waning effect of vaccination, but it is very unlikely that the effect is negative.”

Some smaller remarks:

Abstract

R2.Q14 **Total population data (I think I understand it from the paper (total = all Danish cases during the study period and their household are considered for inclusion), but from the abstract it is unclear what this means).*

AU Thank you for this input. We have now revised so this is clear from the beginning.

R2.Q15 **Specify vaccine effectiveness (full schedule/partial vaccination)*

AU We have now specified these categories throughout the manuscript.

R2.Q16 **Should the difference in Ct-values be reported in the abstract given only 'one third of sd'*

AU We find that vaccinations not only protect against infection, but also against infectiousness. We find that a lower viral load (higher Ct values) is a plausible biological mechanism for this result on infectiousness. A 1.6 point lower Ct value for unvaccinated secondary cases corresponds to a 3 times higher viral load ($2^{1.6}=3$). This finding is both statistically and biologically significant. We believe this to be a main finding of our paper and have re-written the abstract and results to help the reader interpret the result.

Introduction

R2.Q17 35* *“difficult to estimate VE because of prioritized vaccination and waning”. This is true, but the link ('it is therefore essential') with VES and VET is unclear.*

AU We have now rewritten this part.

R2.Q18 52* *“individuals may change their behavior due to the vaccination status of themselves” Is there any indication that this is true for households?*

AU Appendix Figure S8 shows the probability of being tested after exposure are different for vaccinated and unvaccinated households contacts, implying a differential testing behavior of household contacts. Also, this is a potential differential bias that we cannot fully control for.

Methods

R2.Q19 *Why was this period chosen?*

AU We start the study period, when Delta began in Denmark. We use whole genome sequencing (WGS) to identify the variant of the primary case. We end the study period, when booster vaccinations are starting to be administered and before Omicron came to Denmark. We have now clarified this in the manuscript.

R2.Q20 *Some sort of attrition table would be nice (loss because of partial vaccination, alpha-infection etc.)*

AU Please see the response to R2.Q6.

R2.Q21 *Uncertainty estimation*

Was the GLM fitted using ML or Quasi ML?

AU The GLM was fit using maximum likelihood

(https://documentation.sas.com/doc/en/pgmsascdc/9.4_3.3/statug/statug_genmod_overview07.htm) – we have now noted this in the manuscript.

R2.Q22 *Different age groups are mentioned:*

109: 5-year age groups

660: 10-year age groups

AU Thank you for noting this. We have now changed to 10-year age groups throughout the manuscript.

Results

R2.Q23 *170: rounding error => 51%*

AU Corrected.

R2.Q24 Table 1: 1380+2947 potential secondary cases in households of 6, why is this number not a multiple of 5? The elimination of partially vaccinated?

AU Yes. That is the reason.

R2.Q25 240: Therefor

AU Corrected.

*R2 Best Regards,
Toon Braeye*

Reviewer #3 (Remarks to the Author):

R3 *This article uses household testing data from Denmark to assess vaccine effectiveness for reducing susceptibility to infection and for reducing infectiousness given infection. These collectively impact transmission potential and are important parameters for mathematical models and guide policy decisions. The data are collected during the Delta era from a large number of households. The data themselves are interesting, but I have some concerns about how they are analyzed and communicated. I was also distracted by the terminology the authors use to describe these effects, and I would strongly prefer they used more commonly accepted names that have been around since the 1990s.*

AU We are happy that the reviewer finds the results important. We have changed the terminology throughout the paper to make it clearer.

Major comments:

1. Terminology.

R3.Q1 - *It would be better to use the recognized terminology of Halloran et al., which defines VE_I as vaccine effectiveness for infectiousness given infection instead of VE_T (which the authors label as transmissibility). In the existing literature, VE_T is used to represent total vaccine effectiveness (the combined impact of direct VE and indirect VE accrued to a vaccinated individual).*

AU We agree and have changed “transmissibility” to “infectiousness” and VE_T to VE_I for vaccine effectiveness against infectiousness. Similarly, we have changed “combined effect” to “total effect” and VEC to VE_T .

R3.Q2 - *VEC (VE_combined) is not a standard measure. From the Halloran and Preziosi 2003 reference, combined refers to an average SAR across groups, not the combined effect of multiple processes. The description in line 99 is not sufficiently clear (“vaccinated individuals compared to unvaccinated individuals” is too vague). Based on their statistical appendix, VEC would be better described as VE_T or total vaccine effectiveness.*

AU We agree and have changed “combined effect” to “total effect” and VEC to VE_T.

R3.Q3 - *Abstract. Clarify that VE_S is susceptibility to infection by adding in reference to infection.*

AU We have now added this to the manuscript.

R3.Q4 - *Line 41. Transmission here is not clearly defined. (Is it defined unconditionally or conditional upon infection?) Clearer terminology is infectiousness given infection. Alternatively, “from infected vaccinated individuals” and “from infected unvaccinated individuals.”*

AU We agree and have now re-written the introduction to be more clear and concise.

R3.Q5 - *Line 347. “VE against transmissibility among unvaccinated potential secondary cases” is difficult phrasing. The infectiousness is coming from the index cases, and TO unvaccinated household contacts.*

AU We agree and have now re-written the discussion/conclusion to be more concise.

2. Calculation of VE in models.

R3.Q6 - Line 173. Define pooled VEs. Does the pooled estimate model for susceptibility include a term for vaccination status of the index case? Or is this handled only through stratification?

AU The pooled VE_s does not include a dummy for the vaccination status of the index, i.e., we only handle this through the stratification. We follow the methods described in Halloran (2003). We have made the description of the pooled VE_s more clear in the methods section and statistical appendix.

R3.Q7 - In Table 2, the pooled estimate for VE against infectiousness (42%) is not between the two stratified estimates (31% and 10%) but is instead higher. This makes me think of Simpson's paradox and confounding due to clustering of vaccination within households. It makes me wonder how interpretable the primary estimate is.

AU The reviewer is correct that the pooled estimate is higher than either of the stratified estimates for VE infectiousness (also the pooled estimate is the same as the highest estimate for VE susceptibility), and we agree that the explanation is most likely due to confounding of vaccination propensity and baseline infectiousness between households. This underscores the importance of including the stratified analysis within the manuscript, and showing the sensitivity to these choices of models is also an important contribution, as policymakers use real-world estimates of VE in order to choose the optimal mitigation response for the still ongoing pandemic. We have included a brief paragraph in the manuscript to make this clear to the reader.

R3.Q8 - Similarly, why does the pooled VE for susceptibility 61% (59,63%) have an identical point estimate and 95% confidence interval to the stratified not vaccinated group? Can the authors confirm this is not an error?

AU Sometimes estimates are just very close to each other. However, they are not perfectly identical: The pooled VE_s=0.6101 (s.e.=0.0088), whereas the estimate for unvaccinated primary cases is VE_s=0.6121 (s.e.=0.0113).

Minor:

R3.Q9 - *Abstract should include the vaccines in use in Denmark and the time period of the study.*

AU Due to the 150 word limit in Nature Communications, we unfortunately cannot add this information.

R3.Q10 - *Abstract: Meaning of “(one third of a standard deviation)” unclear. “A viral load one third of a standard deviation higher than fully vaccinated individuals”?*

AU We have now re-written the abstract. We have changed the wording to “Furthermore, unvaccinated secondary cases with an infection exhibited a three-fold higher viral load compared to fully vaccinated secondary cases with a breakthrough infection.”.

R3.Q11 - *Abstract: “Our results imply that vaccinations reduce susceptibility” ... mention the Delta variant.*

AU Thank you for this pertinent suggestion. We have now made sure that it is clear that it is the effect of vaccination against infection with the Delta VOC that we estimate.

R3.Q12 - *Introduction. “SARS-CoV-2 pandemic” and “COVID-19 infections” would be better rewritten as “COVID-19 pandemic” and “SARS-CoV-2 infections”, and similar.*

AU We agree. Corrected.

R3.Q13 - *Line 40. Provide context for Harris study. Households in England, time period.*

AU We have re-written the introduction and moved the reference to the Harris study to the discussion.

R3.Q14 - *Line 45. It is unfortunate for this paragraph to not cite any of the broad literature on using household studies to estimate VE_I .*

AU We have now included some references for this (Line 136-146).

R3.Q15 - Line 60. *“with and without vaccination against SARS-CoV-2” should be deleted, as VE is itself a relative measure.*

AU Corrected.

R3.Q16 - Section 3.2. *Description of testing protocols in Denmark in this section would be helpful. They are mentioned later in the discussion, but provide important context on how complete the results might be.*

AU We have now included a brief background paragraph on the Danish testing system (Line 306-311).

R3.Q17 - Line 105. *Unclear what “see below” refers to.*

AU We have re-written the Methods section to be more clear.

R3.Q18 - Table 2 includes helpful notation defining the estimators, but this notation is not itself defined in the main text.

AU Thank you for the suggestion. We have re-written the Methods section to include this and have also included this in the main text.

REVIEWERS' COMMENTS

Reviewer #1 (Remarks to the Author):

The authors have elaborately and adequately answered the comments of the reviewers. I have no further comments.

Reviewer #2 (Remarks to the Author):

Dear Authors,

The main comment remains the higher 'pooled' VE_i and VEs compared to the stratified estimates (not and fully). The authors have now added a section in the discussion explaining this finding, but I believe that it might miss the point. Is it not more likely that the pooled VE_i is higher because VEs is confounded within the estimate? Pooled VE_i is estimated by SAR(V,.) over SAR(N,.). In section S2.3 high correlation is described for the vaccination status of householdmembers. SAR(V,.) will thus mainly include (V,V)-observations and (N,.) will mainly include (N,N)-observations. In the stratified analysis (not and fully) this effect is controlled for by only including one vaccination status of the contact e.g. (V,N and N,N). In the stratified analysis VE_i is thus not confounded by VEs. Given the current methodology one would indeed expect the 'pooled' VE_i to be between the 'not' VE_i and the VE_t.

The pooled estimate is described as 'unconditional on the vaccination status of the primary case', maybe 'unadjusted for the vaccination status of the primary case' would be a more clear description?

The confounding within the pooled estimate also explains why in Figure 1 the VE_i drops from 72 in 0-20 year olds to 14 in 20-40 year olds. In the latter age group (V,.) will often be (V,N) since this an age at which people have young children. Figure 1 is thus confounded by time since vaccination and correlation in vaccination status within households. It makes me wonder how interpretable these estimates are given these two important confounders.

Minor comments:

line 59: Comirnaty

line 70: remove 'for'

line 77: capital 'Implying'

Reviewer #3 (Remarks to the Author):

I am satisfied with the revisions. I have no further comments.

Response to REVIEWERS' COMMENTS

Round 2

Reviewer #1 (Remarks to the Author):

R1: *The authors have elaborately and adequately answered the comments of the reviewers. I have no further comments.*

Reviewer #2 (Remarks to the Author):

R2: *Dear Authors,*

AU: Thank you for taking the time and effort for a very comprehensive review. We truly believe that your comments have improved the manuscript tremendously.

R2.Q1: *The main comment remains the higher 'pooled' VE_i and VE_s compared to the stratified estimates (not and fully). The authors have now added a section in the discussion explaining this finding, but I believe that it might miss the point. Is it not more likely that the pooled VE_i is higher because VE_s is confounded within the estimate? Pooled VE_i is estimated by $SAR(V,.)$ over $SAR(N,.)$. In section S2.3 high correlation is described for the vaccination status of household members. $SAR(V,.)$ will thus mainly include (V,V)-observations and (N,.) will mainly include (N,N)-observations. In the stratified analysis (not and fully) this effect is controlled for by only including one vaccination status of the contact e.g. (V,N and N,N). In the stratified analysis VE_i is thus not confounded by VE_s . Given the current methodology one would indeed expect the 'pooled' VE_i to be between the 'not' VE_i and the VE_t .*

AU: Thank you for this comment, which we agree with.

We have changed the second-to-last paragraph of the discussion from

“Table 2 also illustrates that the pooled estimate for VE against infectiousness (VE_i) is higher than either of the two stratified estimates. This is most likely due to confounding of vaccination propensity and baseline infectiousness between households: if a situation exists whereby households with lower vaccination rates also had lower compliance with other anti-infection measures such as handwashing, then we would expect to see the same relationship as we observe in our data due to Simpson’s paradox. [...]”

to

“From the stratified VE estimates, we find that both the $VE_s > 0$ and $VE_i > 0$. Furthermore, we find a high correlation of vaccination status within households. Combined, this suggests that the pooled VE estimates are confounded, e.g., the pooled VE_i estimate is larger than both of the two stratified VE_i estimates, as it is confounded by the VE_s . [...]”

We have furthermore changed a sentence in the abstract from “[...] In this setting, we estimated VE_s as 61% (95%-CI: 59-63) and VE_i as 42% (95%-CI: 39-45). [...]” to “[...] In this setting, we estimated VE_s as 61% (95%-CI: 59-63), when the primary cases was unvaccinated, and VE_i as 31% (95%-CI: 26-36), when the household contact was unvaccinated. [...]”

R2.Q2: *The pooled estimate is described as 'unconditional on the vaccination status of the primary case', maybe 'unadjusted for the vaccination status of the primary case' would be a more clear description?*

AU: We do not control/adjust for vaccination at any point in the manuscript. We do stratify/condition on vaccination status. Thus, we believe that unconditional is the right terminology here, in order to not confuse the reader.

R2.Q3: The confounding within the pooled estimate also explains why in Figure 1 the VE_i drops from 72 in 0-20 year olds to 14 in 20-40 year olds. In the latter age group (V,_.) will often be (V,N) since this an age at which people have young children. Figure 1 is thus confounded by time since vaccination and correlation in vaccination status within households. It makes me wonder how interpretable these estimates are given these two important confounders.

AU: We agree that the crude VE estimates are clearly confounded by several characteristics, including time since vaccination of the primary case and contact, household size, and the composition of age groups.

We agree, that the (V,_.) \approx (V,N) for primary cases aged 20-40 years, as many of the contacts will be their children. However, we believe the same holds for primary cases aged 0-20 years, as many of the contacts will be their siblings. A major difference between the transmission from primary cases aged 0-20 versus 20-40 years to contacts aged 0-20 years is the that the 0-20 years are primarily siblings to the contacts, whereas the 20-40 years primarily are parents to the contacts. This leads to differentials in the contacts, i.e., it is easier to separate siblings from each other than parents from their child. Thus, these estimates are related to both the behavioral as well as biological effects of transmission.

Finally, we believe the editor must decide on the disagreement between reviewer 1 and 2 on the interpretability and the value added of Figure 1.

R2: Minor comments:

line 59: Comirnaty

line 70: remove 'for'

line 77: capital 'Implying'

AU: Corrected

Reviewer #3 (Remarks to the Author):

R3: *I am satisfied with the revisions. I have no further comments.*